# Nanostructured Poly-l-lactide and Polyglycerol Adipate Carriers for the Encapsulation of Usnic Acid: A Promising Approach for Hepatoprotection

**DOI:** 10.3390/polym16030427

**Published:** 2024-02-03

**Authors:** Benedetta Brugnoli, Greta Perna, Sara Alfano, Antonella Piozzi, Luciano Galantini, Eleni Axioti, Vincenzo Taresco, Alessia Mariano, Anna Scotto d’Abusco, Stefano Vecchio Ciprioti, Iolanda Francolini

**Affiliations:** 1Department of Chemistry, Sapienza University of Rome, P.le A. Moro, 00185 Rome, Italy; benedetta.brugnoli@uniroma1.it (B.B.); perna.1967260@studenti.uniroma1.it (G.P.); sara.alfano@uniroma1.it (S.A.); antonella.piozzi@uniroma1.it (A.P.); luciano.galantini@uniroma1.it (L.G.); 2School of Chemistry, University of Nottingham, Nottingham NG7 2RD, UK; eleni.axioti@nottingham.ac.uk (E.A.); vincenzo.taresco@nottingham.ac.uk (V.T.); 3Department of Biochemical Sciences, Sapienza University of Rome, P.le A. Moro, 5, 00185 Rome, Italy; alessia.mariano@uniroma1.it (A.M.); anna.scottodabusco@uniroma1.it (A.S.d.); 4Department of Basic and Applied Science for Engineering, Sapienza University of Rome, Via del Castro Laurenziano 7, 00161 Rome, Italy

**Keywords:** poly-l-lactide, polyglycerol adipate, usnic acid, drug delivery, polymer nanoparticles, drug hepatoprotection

## Abstract

The present study investigates the utilization of nanoparticles based on poly-l-lactide (PLLA) and polyglycerol adipate (PGA), alone and blended, for the encapsulation of usnic acid (UA), a potent natural compound with various therapeutic properties including antimicrobial and anticancer activities. The development of these carriers offers an innovative approach to overcome the challenges associated with usnic acid’s limited aqueous solubility, bioavailability, and hepatotoxicity. The nanosystems were characterized according to their physicochemical properties (among others, size, zeta potential, thermal properties), apparent aqueous solubility, and in vitro cytotoxicity. Interestingly, the nanocarrier obtained with the PLLA-PGA 50/50 weight ratio blend showed both the lowest size and the highest UA apparent solubility as well as the ability to decrease UA cytotoxicity towards human hepatocytes (HepG2 cells). This research opens new avenues for the effective utilization of these highly degradable and biocompatible PLLA-PGA blends as nanocarriers for reducing the cytotoxicity of usnic acid.

## 1. Introduction

The quest for efficient drug delivery systems has prompted huge interest in the design and development of nanostructured carriers to overcome the challenges associated with the delivery of bioactive compounds. Polymeric nanosystems offer immense potential for drug delivery by providing a versatile platform that can be finely tuned for various therapeutic applications [1,2,3,4]. Their ability to be easily functionalized [5], self-assemble into unique nanostructures [6,7,8], encapsulate and release drugs with controlled kinetics, enhance solubility, and target specific tissues holds promise for revolutionizing the field of medicine, offering solutions to challenges associated with traditional drug administration. Linear aliphatic polyesters, such as polylactide, polyglycolide, or polyhydroxyalkanoates, have garnered considerable attention as promising materials for nanocarriers in drug delivery mainly due to their biocompatibility and biodegradability [9]. In addition, their inherent hydrophobicity makes them good candidates for the incorporation of hydrophobic/low-water-soluble drugs.

The main drawback of this class of polymers is the lack of functional groups that may tune their interaction with drugs as well as provide colloidal stability to the suspension. In addition, most linear aliphatic polyesters possess a high crystallinity degree that may negatively affect drug loading and polymer degradation rate [9,10,11,12]. Among aliphatic polyesters, poly-l-lactide (PLLA) is biocompatible, FDA-approved, and extensively studied for drug delivery [13,14]. One of the drawbacks of PLLA is its hydrophobic nature, which may lead to low loading capacities of hydrophilic drugs and a slow degradation rate. Therefore, PLLA may be not appropriate when rapid drug release or a short-term therapeutic effect is desired [15]. In response to these challenges, the present study delves into the preparation of nanostructured carriers based on PLLA blended with an amorphous, amphiphilic polyester, polyglycerol adipate (PGA), displaying free OH groups in a side chain. The obtained nanocarriers were used for the encapsulation of usnic acid (UA), a secondary metabolite of several lichens extensively studied for its broad variety of biological features, including antimicrobial and anticancer properties [16,17,18].

Despite UA’s promising attributes, the practical application of UA in pharmaceutical and biomedical contexts is hampered by inherent challenges, most notably its limited solubility and bioavailability [16]. In addition, the FDA has reported about 21 cases of serious liver reactions, including hepatic necrosis, fulminant hepatitis, and liver failure, in people who take drugs and dietary supplements containing usnic acid [19,20]. As a result, restrictions have been placed on the intake of UA or products containing UA, strongly limiting their use in therapy. These decisions have prompted studies to reduce UA liver toxicity, still maintaining its pharmacological effects [21]. The encapsulation of UA in suitable polymer nanocarriers has lately shown the potential to decrease drug hepatotoxicity [22,23,24,25,26].

Polyglycerol adipate (PGA) is a glycerol-derived biodegradable amorphous polyester that can be enzymatically synthesized using a lipase enzyme (Novozym 435), known to be selective to primary alcohols, that enables the synthesis of linear polyesters displaying secondary hydroxyl groups [27,28]. Due to its well-balanced amphiphilicity, aided by the secondary hydroxyl groups, PGA is able to self-assemble in water into ca. 100 nm sized nanoparticles giving a stable nanosuspension without the requirement of surfactants [29]. Being an amorphous polymer, in general, PGA has been shown to exhibit improved compatibility with a wider range of pharmaceutical compounds and be highly suited to drug delivery applications [30].

The rationale behind blending PLLA to PGA for the preparation of nanostructures lies in the potential synergistic combination of the individual characteristics of the two polymers. The amorphous component PGA may facilitate drug loading, while the crystalline polymer PLLA may contribute to structural integrity and controlled release, preventing premature drug release and ensuring sustained therapeutic concentrations. In addition, the hydroxyl groups of PGA may contribute to the stabilization of the polymer nanosuspension in water. In order to confirm such hypotheses and study the potential hepato-protection as a consequence of drug encapsulation, in the present work, nanoparticles of PLLA, PGA, and a PLLA-PGA 50/50 blend were prepared by means of nanoprecipitation in water. The size and zeta potential of the obtained particles as well as the stability of the colloidal suspension over time were studied with dynamic light scattering. In order to investigate the compatibility of PLLA and PGA in the 50/50 blend, the thermal properties of the systems were also investigated using differential scanning calorimetry, which is a renowned technique to study the compatibility and stability of polymers, plastic waste, and hybrid organic–inorganic materials [31,32,33,34]. A qualitative degradation assay of NPs was performed using a lipase from porcine pancreas as the degradative enzyme. Usnic acid was then loaded in the nanostructured systems using three UA: (blend)polymer ratios (with total UA content of 1.5, 3, and 6 mg). The apparent water solubility of the systems was determined using UV–vis spectroscopy, and the cytotoxicity of pure and UA-loaded NPs was studied in vitro vs. in human hepatocytes (HepG2 cells). The results evidenced how the blended system, PLLA-PGA, out-performed the others in terms of increasing drug solubility in water and decreasing its cytotoxicity on the HepG2 cell line.

## 2. Materials and Methods

### 2.1. Materials

Lipase immobilized from *Candida antarctica* (>5000 U/g), glycerol (Gly), PLLA (50,000 g/mol), and UA (2,6-diacetyl-7,9-dihydroxy-8,9b-dimethyl-1,3(2H,9bH)-dibenzofurandione) were bought from Sigma-Aldrich, Milan, Italy. Divinyl adipate (DVA) was purchased from TCI America (Portland, OR, USA). All the chemicals were used without further purification. Tetrahydrofuran was acquired from Sigma-Aldrich, Italy.

### 2.2. Methods

#### 2.2.1. Nuclear Magnetic Resonance (NMR)

Polymer formation and repetitive unit chemical structure assignment were determined using ^1^H NMR spectroscopy. Approximately 7 mg of sample was dissolved in 0.7 mL of acetone-d6, analyzed using a Bruker DPX 300 MHz spectrometer (Ettlingen, Germany).

#### 2.2.2. Attenuated Total Reflectance–Fourier-Transform Infrared Spectroscopy (ATR-FTIR)

Infrared spectroscopy was employed to evaluate the success of polymer functionalization. FTIR spectra were acquired in an attenuated total reflection mode (ATR) using a Nicolet 6700 (Thermo Fisher Scientific, Waltham, MA, USA) with a Golden Gate Single Reflection ATR System model equipped with a synthetic diamond having an angle of incidence equal to 45°. The OMNIC 8.3 software was used to process the data obtained during the experiment. Measurements were conducted in a spectral range within 4000–650 cm^−1^, with a resolution of 4 cm^−1^ and 200 scans per spectrum.

#### 2.2.3. Differential Scanning Calorimetry (DSC)

A DSC analysis was performed on 4–5 mg of sample using a Mettler DSC 822 apparatus (Mettler Toledo, Columbus, Ohio, USA) under N_2_ flow (30 mL/min). DSC experiments were acquired at 10 °C min^−1^ in a temperature range from −70 to 190 °C. Two heating cycles were recorded in order to remove any thermal history of the polymers. The second heating cycle, carried out in the same temperature range, was used to determine the glass transition temperature (T_g_), melting temperature (T_m_), and crystallization temperature (T_c_) of the polymers. The degree of crystallinity of the PLLA in the blend was calculated using the following equation:$${∆}_{c}=\frac{{∆H}_{m}-{∆H}_{c}}{{∆H}_{m}^{0}\times W^{PLA}}\times 100$$
where Δ*H_m_* is the enthalpy of melting for PLLA, Δ*H_c_* stands for the enthalpy of crystallization for PLLA, *W^PLA^* is the PLLA weight fraction in the blend system, and ${∆H}_{m}^{0}$ is the heat of fusion for 100% polymer crystal (${∆H}_{m}^{0}$= 93.0 J/g) [35].

#### 2.2.4. Synthesis of Polyglycerol Adipate

Polyglycerol adipate was synthesized as previously described [36]. Enzymatic polymerization was carried out in a glass vial by adding DVA (6.31 mmol, 1.25 g) and Gly (6.31 mmol). Then, THF (10 mL) was added to solubilize the mixture (200 rpm) at 50 °C. After solubilizing, lipase (0.055 g) was added to ensure the polymerization, and the reaction was left reacting for 5 h.

A needle was inserted through the rubber septum in order to facilitate the release of acetaldehyde. The reaction was stopped by simply removing the immobilized enzyme by means of filtration, followed by evaporation of the solvent under reduced pressure. The resultant highly viscous yellow liquid was stored at −20 °C in order to minimize possible hydrolysis side reactions.

#### 2.2.5. Nanoparticles’ Preparation

Polymer nanoparticles were prepared by nanoprecipitation in water, using THF as the solvent for the polymer, as previously reported [34]. The nanoparticles were obtained from pure PLLA and PGA or by blending PGA and PLLA in a 50/50 weight ratio.

PGA, PLLA, or PLLA/PGA 50/50 were dissolved in THF (1 mL) to obtain a 5 mg/mL final concentration. The resulting solution (1 mL) was added dropwise to distilled water (10 mL) under stirring (250 rpm) and left overnight to let the solvent evaporate. To promote PLLA solubilization in THF, PLLA was quenched in liquid nitrogen (−196 °C) to obtain an amorphous material and dissolved in THF after 30 min of sonication.

#### 2.2.6. Usnic Acid Encapsulation in Polymer Nanoparticles

Usnic acid was entrapped in polymer nanoparticles by dissolving different amounts of UA (1.5 mg, 3 mg, and 6 mg) into 1 mL of polymer solution in THF (5 mg/mL). The solution (1 mL) was then added dropwise to distilled water (10 mL) under stirring (250 rpm) and left overnight to let the solvent evaporate. In this way, three final UA concentrations 10, 20, and 40 times higher than the UA toxic concentration towards HepG2 cells (15 μg/mL) were obtained. The samples were named Polymer-UAX where X was 10x 20x, or 40x.

#### 2.2.7. Dynamic Light Scattering (DLS)

The nanoparticles’ hydrodynamic size, polydispersity index (PDI), and ζ-potential of plain or loaded with UA (after filtration, 450 nm) were determined by using a Zetasizer Nano apparatus (Malvern Instruments Ltd., Malvern, UK) equipped with a 4 mW HeNe laser source (632.8 nm). The measurements were carried out at 25 °C. The stability of the suspension was studied over a period of 7 days by measuring the size of the particles over time.

#### 2.2.8. Enzymatic Degradation

Lipase from porcine pancreas, Type II (≥125 units/mg protein (using olive oil (30 min incubation)), 30–90 units/mg protein (using triacetin)) was used in the experiments. A solution of the enzyme at 10 mg/mL in PBS was prepared. A total of 50 µL of this solution was added to 250 µL of NPs (at a concentration of 2.5 mg/mL in water, as mentioned previously). The effect of the enzyme was observed within 0, 1, 3, and 24 h at 25 °C.

#### 2.2.9. Apparent Solubility Test

In order to assess the ability of polymer nanoparticles to encapsulate the drug, UV-vis spectroscopy was employed according to a previously developed method [37]. Specifically, the variation in absorbance between the free drug in water and its polymer formulation was measured. This was achieved by normalizing the absorbance values of the drug−polymer dispersions against the absorbance of the free drug in water at the same maximum wavelength. The resulting values (Δ*A*%), calculated with the formula below, were then used to compare the ability of the different polymer formulations to encapsulate the drug. The absorbance (*A*_0_) of the drugs alone in water was evaluated at 290 nm. In parallel, the absorbance of the aqueous solutions of drug/polymer blends (*A*) was tested using the absorbance of the polymer solutions as a blank. All the absorbance values were kept in the range 0 < *A* < 1, where the Beer−Lambert law can be considered valid and, thus, so can the correlation between absorbance and drug concentration.
$$∆A\%=\;∆A\times 100=\frac{A-A_{0}}{A_{0}}\times 100$$

#### 2.2.10. Study of Drug Release

Drug cumulative release was evaluated using the dynamic dialysis method [38,39] that involves the physical separation of the drug-loaded nanoparticles from the release environment by usage of a dialysis membrane that allows for ease of sampling at periodic intervals. Specifically, a defined aliquot (1 mL) of UA-loaded nanoparticles suspensions (PLLA-UA20X, PGA-UA20X, and PLLA-PGA-UA20X) was inserted in a dialysis bag (CUT OFF 3500), which was then immersed in 15 mL of deionized water. At predefined times (from 15 to 240 min), the release medium was collected, replaced with fresh water, and analyzed by means of UV–Vis spectroscopy at 290 nm.

#### 2.2.11. Cell Culture and MTS Assay

HepG2 cells, a human hepatocarcinoma cell line, obtained from the American Type Culture Collection (HB-8065, ATCC, Rockville, MD, USA) were used as the hepatocytes model [40,41,42]. The cells were grown in Dulbecco’s modified eagle medium low glucose (Sigma-Aldrich), supplemented with 10% fetal bovine serum (FBS) (Gibco, Thermo Fisher Scientific, Waltham, MA, USA) with 1% penicillin/streptomycin, 1% L-glutamine, and 1% sodium pyruvate (Sigma-Aldrich), at 37 °C with 5% CO_2_.

To assess the hepatotoxic effect, an MTS 3-[4,5-dimethylthiazol-2 -yl]-5-[3-carboxymethoxyphenyl]-2-[4-sulfophenyl]-2H-tetrazolium)-based colorimetric assay was performed (Promega Corporation, Madison, WI, USA). A total of 5 × 10^3^ HepG2 cells per well were left untreated (CTL) or treated with 90, 60, 30, 15, 7.5, and 3.75 µg/mL of UA and PLLA-UA20X, PGA-UA20X, PLLA-PGA-UA10X, PLLA-PGA-UA20X, and PLLA-PGA-UA40X nanoparticles containing 30, 15, and 7.5 µg/mL of UA for 24, 48 and 72 h. After each time point, a 100 μL MTS solution was added to the wells. Spectrophotometric absorbance was directly measured at 492 nm after 3 h of incubation using a microplate reader (NB-12-0035, NeBiotech, Holden, MA, USA).

#### 2.2.12. Immunofluorescence Analysis

To visualize cell morphology and actin filaments, immunofluorescence experiments were performed. A total of 30 × 10^3^ cells per well were seeded in eight-well-ibidi plates and cultured for 24 and 72 h in the presence of 30 µg/mL of UA and with PLLA-UA20X, PGA-20X, and PLLA-PGA-UA20X nanoparticles containing 30 µg/mL of UA. At the end of the treatments, the cells were washed in PBS, fixed in 100% ethanol for 15 min, at room temperature, and permeabilized with 0.5% Triton-X 100 in PBS, for 10 min, at room temperature. After blocking with 3% bovine serum albumin (BSA) in PBS for 30 min, at room temperature, the cells were incubated with Phalloidin Alexa Fluor 488 (Immunological Sciences, Rome, Italy) 1:40, for 20 min, at room temperature. The cells were ultimately washed in PBS and incubated with DAPI (Invitrogen, Thermo Fisher Scientific, Waltham, MA, USA) to visualize the nuclei. The images were captured with the optical microscope Leica DM IL LED, using a AF6000 modular Microscope (Leica Microsystem, Milan, Italy).

## 3. Results and Discussion

In this study, nanostructured PLLA and PGA carriers for the encapsulation of UA have been prepared and characterized. The aim of this study was to decrease UA toxicity and increase UA bioavailability through its encapsulation into polymer nanoparticles. Three nanostructured polymer systems were prepared based on pure PLLA, pure PGA, or a PLLA-PGA 50/50 blend. PLLA is a linear, crystalline aliphatic polyester, while PGA is a partially branched, amorphous polyester bearing hydroxyl groups. Both polyesters have been shown to be suitable for drug delivery applications. However, due to its hydrophobic and crystalline nature, PLLA may not be suitable for all types of drugs and shows a slow degradation rate. To overcome these issues, PLLA was blended with PGA in a 50/50 weight ratio to investigate the effect of blending on drug encapsulation, degradation, and cytotoxicity of the polymer system. It was hypothesized that the amorphous component, PGA, may facilitate drug loading and contribute to the stabilization of the polymer nanosuspension in water due to the presence of hydroxyl groups in the polymer side chain. Meanwhile, the crystalline PLLA may contribute to structural integrity.

### 3.1. Polyglycerol Adipate Synthesis

PGA has been produced enzymatically following the well-established literature protocol as shown in Figure 1 [43,44]. From the ^1^H-NMR, the absence of the divinyl peaks at 7.30, 4.87, and 4.59 ppm related to the monomer DVA confirmed the success of polymerization. In the spectrum, the signals of the CH_2_ groups of the adipate unit are at 2.4 e 1.6 ppm, while the glycerol CH is at 4.1 ppm when linked to the free OH group (linear structure 1,3-functionalisation) and at 5.1 and 5.3 when 1,2,3-trisubstituted. These last signals are only traces confirming, as for protocol, the linearity of the macromolecule. The end-group methylene groups are at 3.7 ppm and the CH near the end-group at 3.8 ppm. The signals at 3.5 ppm and 4.4 ppm are related to the CH_2_ bonded to the free OH group or to the adipate ester oxygen, which is related to glycerol esterification in the 1,2 position rather than 1,3. The glycerol CH_2_ groups, either in the main chain or as end groups, are at 4, 4.2 ppm.

### 3.2. FTIR Spectroscopy and Thermal Properties of Polymers Alone and Blended

The plain polymers and the PLLA-PGA blend were characterized using FTIR-ATR (Figure 2A). The PLLA spectrum showed CH stretching of the methyl group in the range 3000–2800 cm^−1^, the ester C=O stretching band at 1748 cm^−1^, while the peaks located at 1450 cm^−1^ and 1080 cm^−1^ were due to C–H bending in the methyl groups and C-O-C stretching, respectively. The PGA spectrum showed the presence of stretching -OH (3700 to 3100 cm^−1^), the stretching of CH groups (3100 and 2800 cm^−1^), and the C=O of the ester bond at 1728 cm^−1^. The PLLA-PGA blend showed the characteristic peaks of the two pure polymers. A lower intensity of the OH band of the PGA in the PLLA-PGA blend could be due to formation of intra- and inter-molecular H bonding with the PLLA.

In order to investigate the compatibility of PLLA and PGA in the 50/50 blend, the thermal properties of the pure polymers and of the blend were investigated via DSC (Figure 2B), similarly to what had been performed in the past [45]. In Table 1, the glass transition temperature (T_g_), variation in specific heat (∆C_p_), crystallization temperature (T_c_), crystallization enthalpy (∆H_c_), melting temperature (T_m_), and melting enthalpy (∆H_m_) of the polymers are reported.

PLLA is a semicrystalline polymer, as shown by the presence of the melting peak at ca. 165 °C in our analyses. Also, the glass transition of the amorphous region was observable at 55 °C. In contrast, PGA showed a sole glass transition at around −32 °C, confirming its amorphous character. The blending of the two polymers slightly changed the thermal properties of each polymer. Particularly, the PGA glass transition temperature increased to −23 °C, while both the crystallization and melting temperatures of PLLA decreased to ca. 76 °C and ca. 166 °C, respectively, suggesting a partial interaction (through dipole–dipole interactions and H-bonding) between the two matrices. The crystallinity of the PLLA in the blend increased from ca. 8 to 25% with the addition of 50% PGA. Therefore, PGA might improve PLA’s crystallization behavior. A similar finding was observed in blends of PLLA with polybutylensuccinate (PBS), where PLLA’s crystallinity increased up to 30% with a low PBS content and decreased for higher PBS contents [46].

### 3.3. Size and Zeta Potential of PLLA, PGA, and PLLA-PGA Nanoparticles

PLLA, PGA, and the PLLA-PGA blend were used to prepare nanocarriers for UA by means of nanoprecipitation in water. In Figure 3A, the DLS curves are shown, while, in Table 2, the size, polydispersity index (PDI), and zeta potential of the obtained nanoparticles are reported.

All the samples showed a monomodal distribution of sizes that ranged from 110 to 150 nm. PLLA and PGA showed similar sizes but slightly different PDI values. The higher PDI observed for PGA may be related to the greater hydrophilicity of this polymer. Among the three systems, the PLLA-PGA nanoparticles possessed the smallest size, which may be a result of nanoaggregates with more tightly assembled polymer chains. This is in agreement with the increase in crystallinity of the PLLA in the blend observed in the DSC experiments. All the samples possessed a negative zeta potential. This is likely due to the presence of free hydroxyl groups in the PGA and terminal COOH groups in the PLLA and the corresponding association of anions to form the outer layers, as shown in case of pegylated nanoparticles [28].

### 3.4. Qualitative Degradation Assay of PLLA, PGA, and PLLA-PGA Nanoparticles

In order to study the behavior of the prepared nanocarriers in the presence of a common lipase, DLS was used as a rapid screening technique, monitoring the change in nanoparticle size after enzyme addition. This experiment was carried out to provide a prediction of degradation behavior of the polymer nanoparticles in the presence of a general lipase rather than a quantitative measure of the degree of degradation. Particularly, the aim of the analysis was to study if blending PLLA with the amorphous component PGA would modify the degradation behavior of PLLA nanoparticles.

Figure 4 shows the change in the nanoparticles size over time as the degradation by lipase progressed.

In all the cases, the nanoparticle size increased during the first 30 min and then decreased with time. In the case of the PLLA nanoparticles, at 2 h of degradation, a monomodal size distribution was observed. The size of these aggregates remained almost constant up to 24 h, suggesting stability of the hydrophobic core. In contrast, the PGA and PLLA-PGA nanoparticles showed more complex DLS traces, with multimodal distributions at all the observed times. In general, the increase in the particle size at the initial degradation may be attributed to an increase in the swelling capacity related to the hydrolyzation of the polyester chains with the consequent formation of shorter and more hydrophilic chains within the aggregates. On the other hand, the reduced hydrophobic character of PGA with increasing degradation time is not sufficient to form stable nanoparticles. Therefore, the formation of nanoparticle populations with a smaller size as the degradation proceeds probably occurs upon the dissociation of PGA chains from the nanoparticles or via the aggregation of the detached chains [47]. Overall, these results indicated that the amorphous PGA, being more affected by lipase degradation, causes disassembling of the initial nanoparticles into aggregates of smaller sizes. The formation of such small aggregates (few nanometers in size) increases the interaction of lipase with the system and, consequently, the degradation rate. Similar results were found by Akagi et al. whilst studying the degradation behavior of amphiphilic graft copolymers consisting of poly(γ-glutamic acid) as the hydrophilic backbone and L-phenylalanine ethylester as the hydrophobic side chain. In this system, with increasing time (up to 72 h), nanoparticles began to decrease in size and, finally, disappeared completely [48].

### 3.5. Encapsulation of UA into PLLA, PGA, and PLLA-PGA Nanoparticles

UA was entrapped in the polymer nanoparticles by being dissolved into the polymer solution, followed by nanoprecipitation in water. The first samples were obtained by dissolving 1.5 mg of the drug into 1 mL of the polymer solution in THF (5 mg/mL) such as to obtain an UA concentration 20x the UA toxic dose. Figure 5A shows the DLS curves of PLLA-UA20X, PGA-UA20X, and PLLA-PGA-UA20X nanoparticles, while Table 3 reports the size, polydispersity index (PDI), and zeta potential of the UA-loaded nanoparticles. PLLA-UA20X and PLLA-PGA-UA20X showed a monomodal distribution of size, contrarily to PGA, where two populations coexisted. PLLA-PGA-UA20X showed a size slightly smaller than PLLA-UA20X and also good stability up to 1 week (Figure 5B).

Therefore, PLLA-PGA was chosen for the encapsulation of UA at two other loading capacities, specifically 10x and 40x. As shown in Table 3, PLLA-PGA-UA20X still showed the best properties in terms of size, PDI, and zeta potential. Indeed, an increase in size and a slight decrease in the zeta potential (and a possible consequent instability in an aqueous environment) value was observed especially for the highest UA-loaded amount (PLLA-PGA-UA40X).

Furthermore, the drug’s apparent solubility in water, determined as described in the Materials and Methods section, improved significantly after encapsulation. The best performing sample was PLLA-PGA-UA20X, for which the UA solubility in water increased up to 800 times compared to the free drug (Figure 5C).

### 3.6. Thermal Properties of UA-Loaded Nanoparticles

In order to investigate the physical state of UA once entrapped in the nanocarriers, a DSC analysis was carried out [49,50]. Indeed, the presence of drug crystallization or melting peaks gives researchers an indication about the state of a drug, amorphous or crystalline, in a formulation. In addition, the thermal behavior of a drug/polymer formulation provides crucial information on the level of interactions between the polymer and the drug within the formulation itself. This information is critical for optimizing the formulation process and ensuring the desired drug release characteristics. Specifically, the impact of the drug’s physical state on drug release has been extensively studied [51,52,53]. Amorphous drugs often exhibit a higher water solubility compared to their crystalline counterparts. The higher solubility and faster dissolution of amorphous drugs can contribute to an improved bioavailability. This is particularly important for poorly water-soluble drugs, like usnic acid 1 mg/mL, since enhanced dissolution can lead to better absorption in the gastrointestinal tract. In contrast, crystalline drug formulations may exhibit more sustained release profiles due to the slower dissolution of the drug from the crystal lattice. This can be advantageous in designing controlled-release or extended-release formulations. At the same time, the state (amorphous or crystalline) of a polymer can also affect the kinetics of drug release [12].

In Figure 5D, the DSC curves of the free UA and of the nanoparticles loaded with UA at a 20x concentration are reported. In Table 4, the thermal properties of the samples are reported.

UA is a crystalline drug showing a sharp melting peak centered at 195 °C and an enthalpy of fusion of 111.4 J/g (Figure 5D). When UA was encapsulated into polymer nanoparticles, its melting peak broadened and was centered at a lower temperature compared to the free drug. A significant decrease in the drug melting temperature from 195 °C to 175–179 °C was observed in presence of PGA, suggesting a preferential interaction of the drug with this amorphous polymer. Also, the lower values of the UA melting enthalpy in the polymer nanoparticles, obtained by normalizing the melting enthalpy for the weight of the drug present in the formulation, compared to the free drug, suggest that UA was partially encapsulated in an amorphous state. The PLLA-PGA-UA20x nanoparticles showed the lowest value of melting enthalpy. Therefore, it is possible to infer that PLLA-PGA-UA20x encapsulated the highest fraction of amorphous UA. This finding may partially explain the high UA apparent water solubility recorded for this system.

### 3.7. Drug Release from nanoparticles

In Figure 6, the cumulative release of usnic acid is reported. Although the differences in the drug release profiles among the three systems are low, it is possible to observe that the PGA-UA20X nanoparticles, obtained with the amorphous polymer PGA, showed the highest released amounts over time, reaching complete drug release at 120 min. Indeed, amorphous polymers are characterized by a disordered molecular structure, which often leads to a higher polymer chain mobility compared to crystalline polymers. When the amorphous polymer comes into contact with a dissolution medium, polymer swelling can occur. Then, the swollen matrix allows for the penetration of the dissolution medium into the polymer, facilitating drug release. Matrix expansion can increase the surface area available for drug dissolution and diffusion, influencing the overall drug release rate. These phenomena have been observed for several amorphous polymers used for drug delivery, including polyvinylpyrrolidone [54] and polylactic-glycolic copolymers [55].

As far as the PLLA-UA20X and PLLA-PGA-UA20X systems are concerned, in the initial hour, they exhibited comparable drug release profiles. However, over extended periods, a disparity in the amounts of drug released became evident. Notably, the PLLA-PGA-UA20X nanoparticles demonstrated a slightly higher release of the drug compared to PLLA-UA20X. It is presumed that the crystalline phase of PLLA within the PLLA-PGA-UA20X formulation contributes to the structural integrity of the system. This, in turn, appears to delay the swelling of the amorphous PGA and the overall drug release.

### 3.8. In Vitro Cytotoxicity Test

In order to evaluate, in vitro, the minimum hepatotoxic concentration of UA, HEPG2 cells were treated with different concentrations of usnic acid, in a range from 90 to 3.75 μg/mL, for 24, 48, and 72 h. As can be observed in Figure 7, under these experimental conditions, UA showed a hepatotoxic effect at 90 and 60 µg/mL at all the analyzed times, decreasing by more than 50% the percentage of living cells compared to the untreated ones. Moreover, a statistically significant decrease was observed after the 72 h treatment in the presence of 30 and 15 μg/mL of UA, while a consistent decrease was observed after 24 h and, to a greater extent, after the 48 h treatment, even if without statistical significance.

In our previous work, we have demonstrated that the incorporation of an UA derivative into chitosan nanoparticles significantly reduces its cytotoxicity [26]. In this study, the effect of UA incorporated in three different nanoparticles prepared by synthetic polyesters PLLA, PGA, and PLLA-PGA on HepG2 cells’ viability has been evaluated. Based on the result obtained in above experiment (Figure 7), 30 and 15 μg/mL UA concentrations were taken into consideration as the lowest analyzed toxic doses and 7.5 μg/mL as the highest non-toxic analyzed concentration. HepG2 cells were, therefore, treated with the corresponding concentrations of the different polymer nanoparticles diluted up to 30, 15, and 7.5 μg/mL UA concentrations. A significant increase in cell viability was observed in all the analyzed conditions and time points, compared to the cells treated with UA alone. The cell viability of the cells treated with polymer nanoparticles containing UA was comparable to the cell viability of the untreated cells (100%, represented by a horizontal black line) (Figure 8).

### 3.9. Morphological Assessment of HepG2 Cells

To evaluate the effects on cytoskeleton morphology of UA compared to UA incorporated in polymer nanoparticles, the treated cells were analyzed by immunostaining the actin with Phalloidin. Based on the results obtained measuring cell viability, we chose the concentration of 30 μg/mL UA and PLLA-UA20X, PGA-UA20X, and PLLA-PGA-UA20X containing 30 μg/mL UA to perform this analysis. As shown in Figure 9, the cytoskeleton of the UA-treated HepG2 cells appeared quite disrupted after 24 h (Figure 9A) and completely disrupted after 72 h (Figure 9B). The nuclei were very small, and the actin filaments were completely lost, suggesting that UA induced necrosis in the cells. The cells treated with UA incorporated in PLLA, PGA, and PLLA-PGA showed a cytoskeleton morphology similar to the untreated cells after 24 h and 72 h (Figure 9). To better highlight the cytoskeleton, the pictures were also acquired at a higher magnification of 63x (Figure 10). At this magnification, the actin filaments were well arranged in all the nanoparticle-treated cells, mainly in the PLLA-PGA-UA20X sample and after 72 h treatment. The loss of cytoskeleton structure was confirmed in the UA-treated cells. The advantages of the administration of UA embedded in polymeric nanoparticles were particularly evident after 72 h treatment. These cells appeared to be well arranged and they formed a complex network, with several cell–cell interactions confirming the protective role of UA incorporation in nanoparticles. Among the three different nanoparticles, the PLLA-PGA-UA20X showed an improved actin structure with respect to the untreated cells and to cells treated with PLLA-UA20X and PGA-UA20X (Figure 10B).

## 4. Conclusions

This study investigated the use of nanostructured PLLA and PGA carriers for the encapsulation of UA. Overall, the FTIR-ATR and DSC results suggested that the PLLA-PGA (50/50) blend is a partially miscible system. The hydroxyl groups in PGA may interact with PLLA through hydrogen bonding. The presence of PGA also increases the crystallinity of PLLA in the blend. The DLS and zeta potential measurements showed that all three types of nanoparticles are well-dispersed and stable in an aqueous solution. Enzymatic degradation was carried out in lipase solution, indicating that the presence of PGA in the blend promotes the degradation of PLLA nanoparticles. All three prepared nanoformulations were able to encapsulate UA effectively. In particular, the PLLA-PGA blend system self-assembled in water, providing nanoparticles with smaller sizes than those of pure polymers. After encapsulation in PLLA-PGA, the apparent solubility of UA increased up to 800 times compared to the free drug. The DSC measurements evidenced that UA was partially encapsulated in an amorphous state; this may contribute to the observed UA increased solubility. The cell viability of cells treated with UA-loaded nanoparticles was found to be comparable to the cell viability of the untreated cells. Therefore, a significant decrease in cytotoxicity was observed in all the analyzed polymer formulations, compared to the cells treated with free UA. The observation of cell cytoskeleton morphology evidenced that the cells treated with UA-loaded nanoparticles were well arranged and formed a complex network, with several cell–cell interactions confirming the protective role of UA incorporation in nanoparticles. Overall, encapsulating UA in PLLA-PGA nanoparticles can significantly reduce its cytotoxic effects on the HepG2 cell line and protect the cytoskeleton.

## Figures and Tables

**Figure 1 polymers-16-00427-f001:**
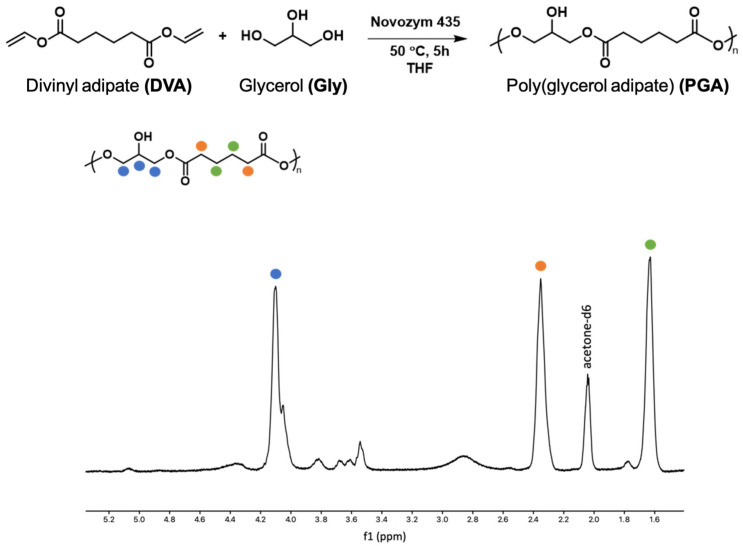
Scheme of PGA polymerization and ^1^H-NMR of PGA.

**Figure 2 polymers-16-00427-f002:**
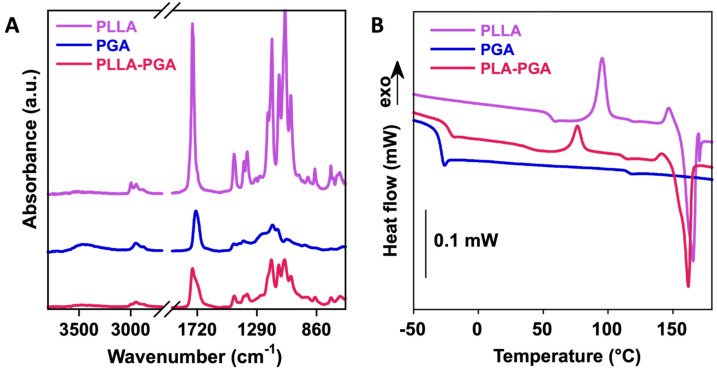
FTIR-ATR (**A**) and DSC curves (**B**) of PLLA, PGA, and PLLA-PGA blend.

**Figure 3 polymers-16-00427-f003:**
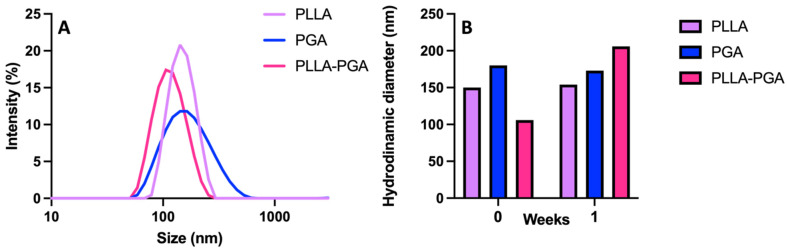
DLS curves of PLLA, PGA, and PLLA-PGA blend (**A**); stability of the nanoparticles determined by monitoring particle size at 1 week (**B**).

**Figure 4 polymers-16-00427-f004:**
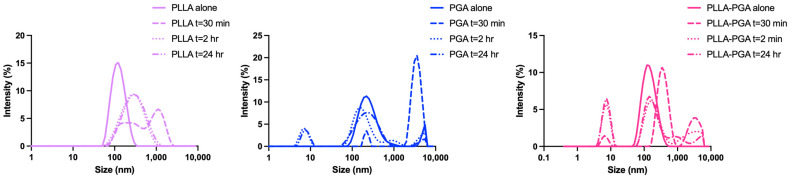
Evolution in size of PLLA, PGA, and PLLA-PGA nanoparticles over time after the addition of lipase (10 mg/mL).

**Figure 5 polymers-16-00427-f005:**
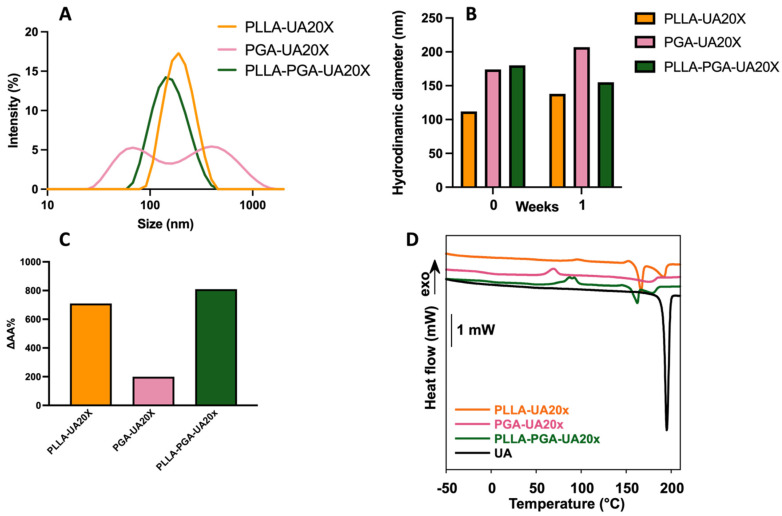
DLS curves of UA-loaded nanoparticles at a 20x UA concentration (**A**); stability of the UA-loaded nanoparticles at 1 week (**B**); UA apparent solubility as determined via UV–vis spectroscopy (**C**); and DSC curves of free UA and UA-loaded nanoparticles at a 20x UA concentration (**D**). In the inset of panel D, a magnification of the thermogram from 50 to 210 °C is reported to show the peaks of crystallization of PLLA and UA.

**Figure 6 polymers-16-00427-f006:**
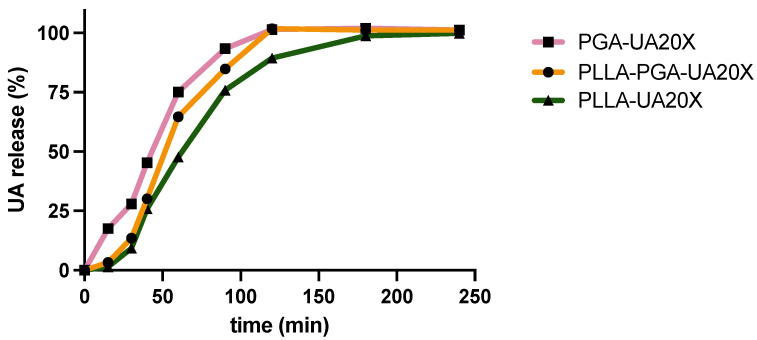
Usnic acid release over time from PLLA-20X, PGA-20X, and PLLA-PGA20X nanoparticles.

**Figure 7 polymers-16-00427-f007:**
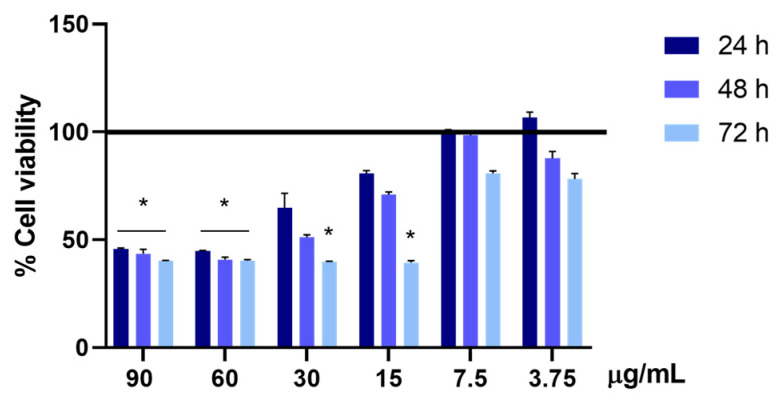
Cell viability is assessed using the MTS assay. The viability of the HepG2 cells treated with 90, 60, 30, 15, 7.5, and 3.75 µg/mL of UA is evaluated after 24, 48, and 72 h of treatment. The cell viability of the samples is normalized to that of the untreated cells, which is reported to be 100% and represented by a horizontal black line. The results are expressed as the mean ± standard deviation of the data obtained by means of three different experiments. Statistical significance is * *p* < 0.05 vs. untreated cells.

**Figure 8 polymers-16-00427-f008:**
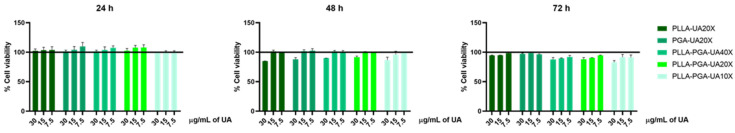
Cell viability is assessed via the MTS assay. The viability of the HepG2 cells treated with PLLA-UA20X, PGA-UA20X, and PLLA-PGA-UA10X, 20X, and 40X containing 30, 15, and 7.5 µg/mL of UA is evaluated after 24, 48, and 72 h of treatment. The cell viability of the samples is normalized to that of the untreated cells, which is reported to be 100% and represented by a horizontal black line. The results are expressed as the mean ± standard deviation of the data obtained by means of three different experiments.

**Figure 9 polymers-16-00427-f009:**
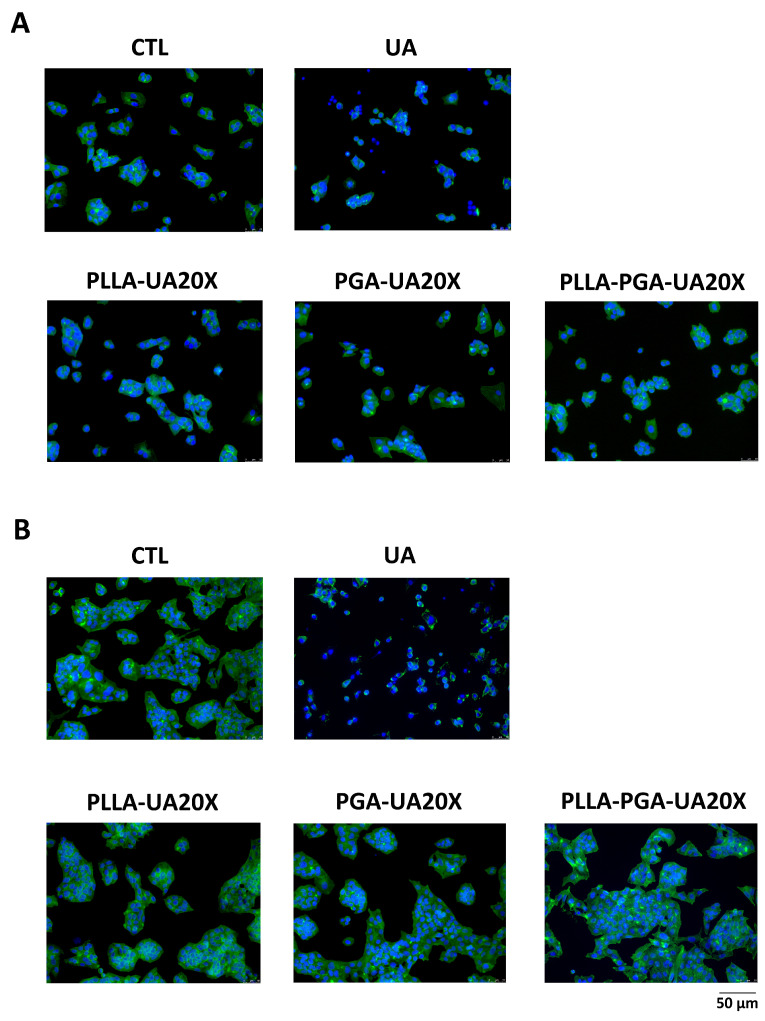
HepG2 cytoskeleton morphology analyses after treatment with UA and UA embedded in polymeric nanoparticles. The cells are treated with UA (30 μg/mL) and PLLA-UA20X, PGA-UA20X, and PLLA-PGA-UA20x containing 30 μg/mL UA for 24 h (**A**) and 72 h (**B**) and then analyzed via an immunofluorescence assay using Phalloidin Alexa Fluor 488 to highlight the actin filaments. The nuclei are stained with DAPI (original magnification 20x, scale bar = 50 μm).

**Figure 10 polymers-16-00427-f010:**
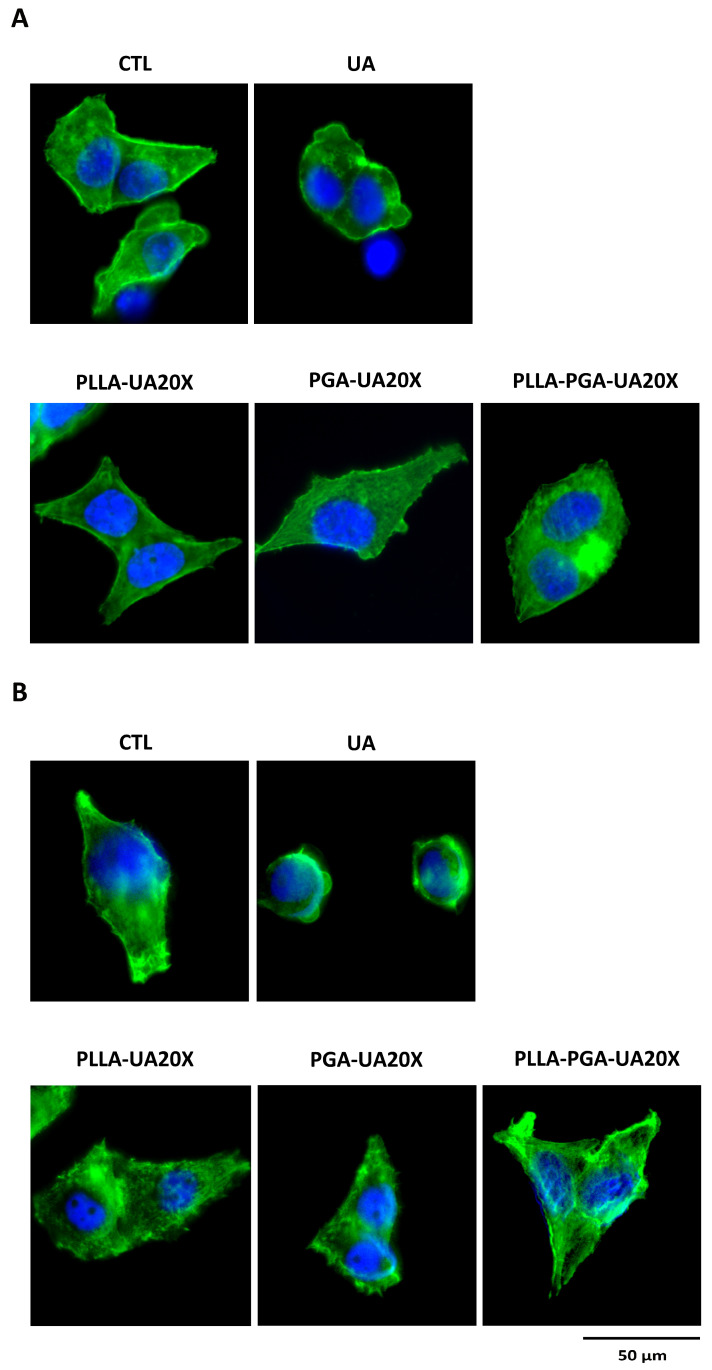
HepG2 cytoskeleton morphology analyses after treatment with UA and UA embedded in polymeric nanoparticles. The cells are treated with UA (30 μg/mL) and PLLA-UA20X, PGA-UA20X, and PLLA-PGA-UA20x containing 30 μg/mL UA for 24 h (**A**) and 72 h (**B**) and then analyzed via an immunofluorescence assay using Phalloidin Alexa Fluor 488 to highlight the actin filaments. The nuclei are stained with DAPI (original magnification 63x, scale bar = 50 μm).

**Table 1 polymers-16-00427-t001:** Thermal properties of polymers obtained using DSC: glass transition temperature (T_g_), variation in specific heat (∆C_p_), crystallization temperature (Tc), crystallization enthalpy (∆H_c_) melting temperature (T_m_), melting enthalpy (∆H_m_), and degree of crystallization ($\chi_{c}$).

Sample	T_g_ (°C)	∆c_p_ (J/g)	T_c_ (°C)	∆H_c_ (J/g)	T_m_ (°C)	∆H_m_(J/g)	$\chi_{c}$ (%)
PLLA	55	0.16	95	9.51	166	16.61	7.6
PGA	−32	0.55	-	-	-	-	-
PLLA-PGA	−23.4 (PGA) 47 (PLLA)	0.25 0.02	76	7.88	162	31.72	25.6

**Table 2 polymers-16-00427-t002:** Size, polydispersity index (PDI), and zeta potential of the PLLA, PGA, and PLLA-PGA nanoparticles, obtained by means of DLS cumulant analysis. Data represent the mean ± standard deviation on three series of repeated measurements.

Sample	Hydrodynamic Diameter (nm)	PDI	ς-Potential (mW)
PLLA	140.0 ± 2.0	0.07 ± 0.01	−34.0 ± 1.0
PGA	150.0 ± 1.0	0.19 ± 0.02	−13.0 ± 2.0
PLLA-PGA	111.0 ± 2.0	0.14 ± 0.03	−24.0 ± 2.0

**Table 3 polymers-16-00427-t003:** Size, polydispersity index (PDI), and zeta potential of PLLA, PGA, and PLLA-PGA nanoparticles loaded with different amounts of UA, obtained via DLS cumulant analysis. Data represent the mean ± standard deviation on three series of repeated measurements.

Sample	Hydrodynamic Diameter (nm)	PDI	ς-Potential (mW)
PLLA-UA20X	188.0 ± 3.0	0.15 ± 0.02	−25.0 ± 1.0
PGA-UA-20X	160.0 ± 20.0	0.40 ± 0.10	−21.0 ± 8.0
PLLA-PGA-UA20X	180.0 ± 20.0	0.30 ± 0.10	−23.0 ± 3.0
PLLA-PGA-UA10X	210.0 ± 20.0	0.29 ± 0.04	−18.0 ± 3.0
PLLA-PGA-UA40X	270.0 ± 40.0	0.37 ± 0.05	−19.0 ± 2.0

**Table 4 polymers-16-00427-t004:** Thermal properties of free UA and of UA-loaded nanoparticles as determined by means of DSC: glass transition temperature (T_g_), variation in specific heat (∆C_p_), crystallization temperature (Tc), crystallization enthalpy (∆H_c_), melting temperature (T_m_), and melting enthalpy (∆H_m_).

Sample	T_g_ (°C)	∆c_p_ (J/g)	T_c_ (°C)	∆H_c_ (J/g)	T_m_ (°C)	∆H_m_ (J/g)	$\chi_{c}$ (%)	T_c_ (°C)	∆H_c_ (J/g)	T_m_ (°C)	∆H_m_ (J/g)
	POLYMER							DRUG			
UA	-	-	-	-	-	-	-	-	-	195	111.4
PLLA-UA20X	−53	0.17	97	27,04	167	33.3	6.7	-	-	192	50.2
PGA-UA20X	−14	0.40	-	-	-	-	-	70	34.5	175	56.8
PLLA-PGA-UA20X			ND *	ND	162	36.3	ND	ND	ND	179	21.2

* ND = not determinable because of different overlapping processes.

## Data Availability

Data are contained within the article.

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
