# Peer review of "Nanostructured Poly-l-lactide and Polyglycerol Adipate Carriers for the Encapsulation of Usnic Acid: A Promising Approach for Hepatoprotection"

_polymers, 2024, doi:10.3390/polym16030427_

Round 1
Reviewer 1 Report
Comments and Suggestions for Authors
Benedetta Brugnoli et al reported the development and characterization of nanostructured carriers based on poly-l-lactide (PLLA) and polyglycerol adipate (PGA). for the encapsulation of (UA). The nanocarriers were employed for encapsulation of usnic acid, and the cytotoxicity was determined. However, I think this manuscript needs major revision and is not suitable for publishing in Polymers in its current form. Here are some comments of this manuscript for authors.
1. There are too many paragraphs in the introduction section, please re-organize this section to 4-5 paragraphs.
2. Why did the authors investigate the thermal property of UA-loaded nanoparticles?
3. HepG2 cell is a liver cancer cell line, the authors should use a normal liver cell line to evaluate the hepatotoxicity of UA.
4. The cell line should be written as HepG2 rather than HEPG2.
5. What do the PLLA-UA20X, PGA-UA20X and PLLA-PGA-UA10X, 20X and 40X mean?
6. The changes in the cytoskeleton morphology after UA treatment are not evident, the authors should observe cells under high magnification.
7. The in vitro release behavior of UA from different nanocarriers should be investigated.
8. There are some typos in the manuscript, such as “citotoxicity” in line 26, “put” in line 302. Please check them carefully.
9. The English language throughout the manuscript should be carefully checked and improved by a native English speaker.
Comments on the Quality of English LanguageThe English language throughout the manuscript should be carefully checked and improved by a native English speaker.
Author Response
see the attached doc file

Reviewer 2 Report
Comments and Suggestions for Authors
Line 2. «Nanostructured». Here and further everywhere in the text, the authors nowhere provide data on the nanostructure of the nanoparticles obtained. Indeed, since the authors use also amorphous polymer, it would be very difficult to establish the features of the spatial structure of the obtained particles in the nanoscale. All the given data indicate only that nanoparticles are formed from the used materials, but no data on the nanostructure of the nanoparticles were not obtained by the authors. I mean the arrangement of polymer chains relative to each other, coordination of usninic acid with functional groups of polymers and how it is all arranged relative to each other in space.
Lines 27-29. «This research opens new avenues for the effective utilization of these highly degradable and biocompatible PLLA-PGA blended nanocarriers for hepatoprotection of usnic acid.» Since, as we know, usnic acid has no liver, hepatoprotection of usnic acid is not possible. It is necessary to reformulate this statement.
Lines 75, 76. «pendant hydroxyl groups» I think it's an incorrect expression especially in a chemical context. One could say that the polymer contains hydroxyl groups, or better to note that "the secondary alcoholic hydroxyl groups remain unsubstituted in the polymer".
Lines 99-100. «hepatotoxicity of pure and UA-loaded NPs was studied in vitro vs. human hepatocytes (HEPG2)». Perhaps it would be more correct to write that in vitro cytotoxicity was investigated, since hepatotoxicity implies toxicological studies on the organ or organism scale and is accompanied by histological studies of the organ (liver), as well as blood studies on the activity of liver-associated enzymes, bilirubin, etc.
Lines 117-118. «Infrared spectroscopy was employed to evaluate the success of polymer functionalization. FTIR spectra were acquired in attenuated total reflection mode (ATR)». Only this specified method of infrared spectroscopy is not suitable for the task set by the authors. As a result of the study of the signal obtained by reflection only a thin surface layer of the material is investigated, not the whole sample. This method is suitable for surface studies, finish coatings, surface chemistry studies, but not for qualitative analyses of the material in mass. It is necessary to prepare a polymer tablet with or without potassium bromide and examine the sample in transmission spectroscopy mode to obtain information about the polymer composition.
Line 148. «were prepared by nanoprecipitation in water» Incorrect statement. If we strictly estimate the rate of sedimentation of polymer nanoparticles in water, this process may take years, if it is possible at all. Indeed, for example, deposition of silica gel particles of 1.5 µm in a 10 cm water layer can take more than a day. In fact, the authors transferred polymers from dissolved to undissolved dispersed state (from tetrahydrofuran to water).
Line 176. «the increased aqueous solubility of the encapsulated drug». It seems to me that it is necessary to rephrase the statements in this paragraph. The authors disperse the active substance in water by encapsulating it in polymer particles. The authors obtain a suspension, not a solution. Thus, we should not talk about the solubility of usninic acid in water.
Figure 1. It is necessary to give data on the integration of signals in the spectrum, including signals not attributed by the authors to the product (5.1, 4.4, 3.9-3.4 p.p.m.)
The most important observation relates to the methodology of the study and the conclusions drawn. The authors obtained polymer particles dispersible in water and loaded with usninic acid. Low cytotoxicity of the obtained particles against hepatocyte cell line was shown. But it is impossible to conclude on the basis of this low hepatotoxicity of nanoparticles, since such a study should be conducted on an organ or organism
Based on the data presented by the authors, it cannot be concluded that nanoparticles release usnic acid. And if usnic acid is not released from nanoparticles, then one cannot speculate about the prospects of using such nanoparticles as carriers of usnic acid. Indeed, for such a conclusion it is necessary to show that usnic acid is released from nanoparticles and has some effect specific to usnic acid. For example, it would be possible to treat a culture of fungal or bacterial cells with the resulting nanoparticles and make sure that usnic acid inhibits their reproduction.
Author Response
see the attached doc file

Round 2
Reviewer 1 Report
Comments and Suggestions for Authors
The authors addresssed my concern.
Comments on the Quality of English Languagenone